# WavJEPA: Semantic learning unlocks robust audio foundation models for raw waveforms

## Abstract

Learning audio representations from raw waveforms overcomes key limitations of spectrogram-based audio representation learning, such as the long latency of spectrogram computation and the loss of phase information. Yet, while self-supervised speech representation learning from raw waveforms has been remarkably successful, these approaches have not achieved similar feats for general-purpose audio representation learning from waveforms. Here, we propose WavJEPA, a waveform-based version of the Joint-Embedding Predictive Architecture. WavJEPA leverages high-level semantic representation learning to tackle the shortcomings of representation learning at the speech unit or token level. We show that this approach substantially outperforms state-of-the-art time-domain audio foundation models across a wide variety of downstream benchmark tasks, while requiring considerably fewer computational resources. Additionally, to overcome the performance drop that time-domain models typically exhibit in noisy and reverberant real-world acoustic environments, we present WavJEPA-Nat. WavJEPA-Nat is a multi-channel extension of the WavJEPA architecture trained on simulated naturalistic scenes. We find that WavJEPA-Nat is highly robust to reverberation and noise. These results highlight the feasibility and computational efficiency of general-purpose audio representation learning from raw waveforms, showcasing the potential for low-latency, robust time-domain audio foundation models for real-world applications.

## 1 Introduction

State-of-the-art approaches for self-supervised general-purpose audio representation learning predominantly operate on spectrograms, that is, time-frequency representations of sound clips (Turian et al., 2022; Yadav et al., 2024; Chen et al., 2023; Gong et al., 2022; Yadav & Tan, 2024). However, these approaches suffer from two fundamental limitations: The latency introduced by the short-time Fourier transform (STFT) required for spectrogram computation impedes real-time deployment (Luo & Mesgarani, 2019), and (2) the loss of phase information reduces the performance on generative audio tasks (Luo & Mesgarani, 2019; Li et al., 2025). In contrast, time-domain models, which learn directly from raw audio waveforms, achieved remarkable success in speech representation learning (Baevski et al., 2020; Hsu et al., 2021; Chen et al., 2021). Crucially, end-to-end audio representation learning from raw waveforms overcomes the key limitations of spectrogram-based audio representation learning (long latency and loss of phase information) (Luo & Mesgarani, 2019). Yet, when state-of-the-art approaches for speech representation learning are trained for general-purpose audio representation learning, their performance is less strong (La Quatra et al., 2024). Furthermore, existing time-domain models exhibit significant degradation in noisy and reverberant acoustic environments compared to their spectrogram-based counterparts, limiting their effectiveness for real-world applications (Yuksel et al., 2025).

To improve these shortcomings, we propose WavJEPA, a novel framework for end-to-end general-purpose audio representation learning from raw waveforms. The idea behind WavJEPA is that the semantic learning capabilities of joint-embedding predictive architectures (JEPAs) (Bardes et al., 2024; Assran et al., 2023; LeCun, 2022) can be leveraged to overcome the limitations of learning representations at the token or speech unit level, which is the typical approach of audio foundation models operating on raw waveforms. Instead, WavJEPA learns semantic representations by predict-

ing the latent representations of training targets from a temporally distributed context representation of the same sound wave.

WavJEPA is the first framework applying semantic learning to general-purpose audio representations in the time domain, surpassing state-of-the-art time-domain approaches on the HEAR (Turian et al., 2022) and ARCH (La Quatra et al., 2024) benchmark suites while requiring only a fraction of the computational resources. Additionally, we address the degraded performance of time-domain models in real-world sound scenes with WavJEPA-Nat, a multi-channel extension of the WavJEPA framework trained on simulated real-world sound scenes. Evaluation on Nat-HEAR (Yuksel et al., 2025), a naturalistic version of the HEAR benchmark suite, demonstrates that WavJEPA-Nat exceeds the robustness of other time-domain foundation models to noise and reverberation. We furthermore elucidate the critical factors for semantic representation learning from raw waveforms through extensive ablation studies, targeting context-target sampling, top-$K$ averaging and the optimal ratio between real-world scenes and dry sound clips. In sum, WavJEPA and WavJEPA-NAT demonstrate that robust time-domain approaches for audio representation learning are feasible and efficient, opening the door to low-latency audio foundation models for real-world applications.

## 2 RELATED WORK

**Spectrogram-based audio representation learning:** These approaches aim to learn general-purpose representations from the time-frequency representation of a sound clip (spectrogram) calculated with a short-time Fourier transform. Masked auto-encoder (MAE) approaches achieve state-of-the-art performance on benchmark suites, learning rich audio representations by reconstructing masked spectrogram patches (He et al., 2022; Yadav et al., 2024; Gong et al., 2022; Chong et al., 2023; Baade et al., 2022; Huang et al., 2022; Yadav & Tan, 2024). Other approaches – inspired by the success in the visual domain (Grill et al., 2020; Bardes et al., 2024; Assran et al., 2023) – avoid reconstructing the original spectrogram input space, instead predicting targets in the latent space (Niizumi et al., 2023; Fei et al., 2024; Chen et al., 2023; 2024).

**Waveform-based audio representation learning:** Representation learning from raw waveforms is based on predictive or contrastive self-supervised learning strategies at the token or speech unit level (Baevski et al., 2020; Hsu et al., 2021; Chen et al., 2021). More recently, Data2Vec (Baevski et al., 2022) introduced a modality-agnostic framework for training across speech, vision, and text domains, leveraging a teacher-student approach. They demonstrated that the proposed latent prediction framework achieves state-of-the art on speech recognition with minimal fine-tuning. While these frameworks have proven extremely fruitful for speech representation learning, they have been less successful in learning general-purpose audio representations (Yadav et al., 2024; Turian et al., 2022; Yuksel et al., 2025; La Quatra et al., 2024).

**Representation learning with joint embedding predictive architectures (JEPAs):** Recent work demonstrated that JEPA models efficiently learn semantic image representations by predicting latent representations of parts of the input image (that is, training targets) from a context representation of other parts of that same image (Assran et al., 2023; Bardes et al., 2024). Based on this success, others applied JEPA models to spectrograms (Fei et al., 2024), EEG signals (Guetschel et al., 2024) and fMRI measurements (Dong et al., 2024), highlighting the versatility of the JEPA framework.

## 3 METHODOLOGY

### 3.1 THE WAVJEPA FRAMEWORK

Our proposed architecture and approach for learning general-purpose audio representations from raw waveforms are illustrated in Figure 1. The WavJEPA framework comprises a waveform encoder, context encoder, target encoder and a predictor. WavJEPA's objective is to predict latent representation of various targets blocks based on a single context block extracted from the same sound wave. As waveform encoder, we use the feature encoder of Wav2Vec 2.0, which is composed of stacked temporal convolution layers (Baevski et al., 2020). Similar to the original I-JEPA architecture (Assran et al., 2023), a Vision Transformer (ViT) (Dosovitskiy et al., 2021) is used for the target encoder, context encoder and predictor. Detailed specifications of the framework components

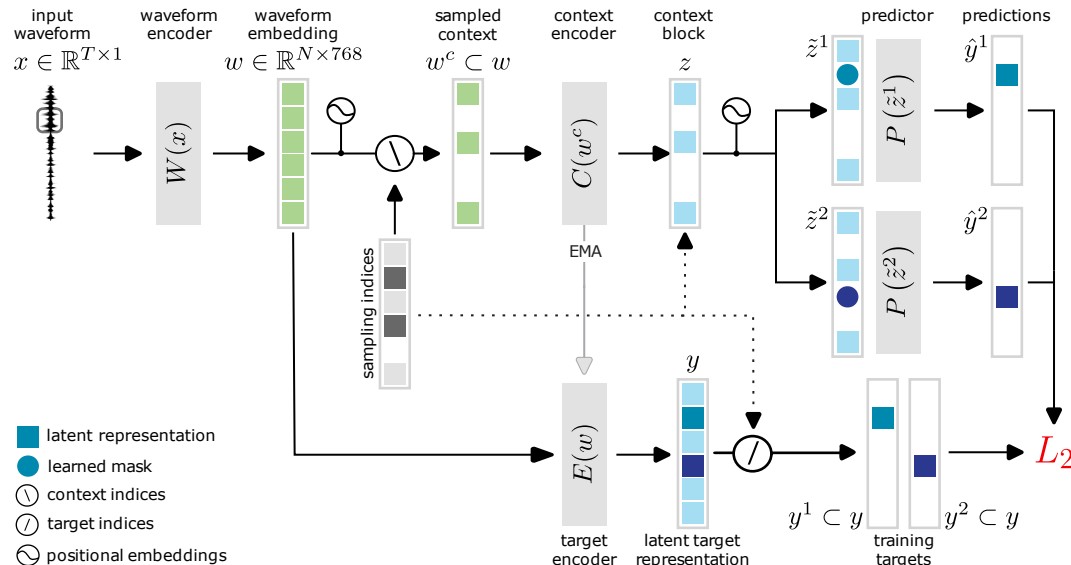

Figure 1: **Semantic representation learning from raw waveforms.** WavJEPA predicts latent target representations at specific locations from a context representation. The weights of the target encoder are not trained but updated using the exponential moving average (EMA) of the weights of the contextencoder.

can be found in Appendix A. In the following, we describe the main components of the WavJEPA framework.

**Waveform encoder:** A sound wave $x \in \mathbb{R}^{T \times 1}$ is transformed into an embedding $w \in \mathbb{R}^{N \times 768}$ by the waveform encoder $w := W(x)$. To obtain a more fine-grained embedding, we removed the last convolutional layer of the Wav2Vec2.0 feature encoder.

**Sampling the context block and target blocks:** A temporally distributed context and $K_{target}$ target blocks are sampled from the $N$ indices in the waveform embedding $w$ in an iterative procedure. We first randomly sample starting indices for the context block with uniform probability $p_{context}$ over the range $[1 \dots N]$. For each starting index, we then include the subsequent $M_{context}$-many indices in our context block. Then, for each target block $k \in [1 \dots K_{target}]$, we randomly sample a starting index and select the subsequent $M_{target}$ indices as training targets. Context indices that overlap with training targets are removed. We repeat this procedure until at least 10% of indices in $[1 \dots N]$ are designated as the context. Ultimately, we obtain $n$ non-overlapping context indices $c_1, \dots, c_n \in [1 \dots N]$, and, for each target block $k \in [1 \dots K_{target}]$, we obtain $M_{target}$ target indices $t_1^k, \dots, t_{M_{target}}^k \in [1 \dots N]$.

**Context encoder:** To obtain a latent context representation $z = \{z_1, \dots, z_n\}$, the context encoder $C(\cdot)$ converts the context waveform embedding $w^c = \{w_{c_1}, \dots, w_{c_n}\}$ into a latent representation $z := C(w^c)$. Attention masking is used to ensure that the context encoder operates only on the context indices $c_1, \dots, c_n$ for the generation of $z$.

**Predictor:** For each target $k \in [1 \dots K_{target}]$, we concatenate the latent context representation $z$ with learnable mask embeddings and additive positional embedding in order to replace the target indices: $\tilde{z}^k = \{z_1, \dots, z_n, m_{t_1^k}, \dots, m_{t_{M_{target}}^k}\}$. The predictor $P(\cdot)$ then takes this augmented latent representation $\tilde{z}^k$ to predict the latent target representations $\hat{y}^k = \{\hat{y}_1^k, \dots, \hat{y}_{M_{target}}^k\}$ such that $\hat{y}^k := P(\tilde{z}^k)$. The predictor is thus applied $K_{target}$ times.

**Target encoder and learning objective:** In this waveform-based approach, latent representations of the sound wave embeddings constitute the targets. The target encoder $E(\cdot)$ converts the whole waveform embedding $w$ into a latent target representation. Similar to Baevski et al. (2022; 2020), the outputs of the top $K$ layers are instance-normalized (Ulyanov et al., 2017) and averaged. For each time step $i \in [1 \dots N]$, we obtain a target embedding $y_i \in \mathbb{R}^{768}$. For each target block

$k \in [1 \ldots K_{target}]$, we select the tokens $y^k = \{y_{t_1^k}, \ldots, y_{t_{M_{target}}^k}\}$ corresponding to the target block indices $t_1^k, \ldots, t_{M_{target}}^k$ and compute the $L_2$ distance between the predicted target representation $\hat{y}^k$ and the actual training target $y^k$. The final loss corresponds to the average error across targets.

**Target encoder parametrization:** The parameters $\Delta$ of the target encoder are not trained, but instead updated on every iteration by an exponential moving average (EMA) of context encoder parameters $\theta$ according to $\Delta \leftarrow \tau\Delta + (1 - \tau)\,\theta$. Here, $\tau$ linearly increased over the first $\tau_n$ updates from $\tau_0$ to target $\tau_e$, after which it was kept constant for the remainder of training.

## 3.2 EXPERIMENTAL SET-UP WAVJEPA

**Data and sound wave embeddings:** We train WavJEPA on the unbalanced training set of AudioSet, which consists of 1.74 million 10-second sound clips scraped from YouTube (Gemmeke et al., 2017). Each sound clip was resampled to 16 kHz and mean centered to enforce equal loudness across sound clips. We then randomly sampled 8 sections of 2 s from each sound clip, effectively increasing the batch size by a factor of 8 in a computationally efficient manner. Finally, each instance is instance normalized (Ulyanov et al., 2017). The waveform encoder converts each 2 s instance into an embedding $w^{200 \times 768}$, effectively resampling the audio to 100 Hz with a stride of 10 ms and a receptive field size of 12.5 ms.

**Pre-training:** We sampled starting indices for the context block with $p = 0.065$ and for target blocks with $p = 0.025$. We set $M$ to 10 for both context block and target block . To update the target encoder parameters $\Delta$, we linearly increased $\tau$ from $\tau_0 = 0.999$ to $\tau_e = 0.99999$ over the first 100,000 steps, after which $\tau$ was kept constant. We used $K = 8$ for the top $K$ averaging.

We trained WavJEPA for 375,000 steps using a batch size of 32 on two NVIDIA H100 94 GB GPUs. Given our in-batch sampling factor of 8, we boost our effective batch size to 256. We use the AdamW optimizer (Loshchilov & Hutter, 2019) with a weight decay coefficient $\lambda_w = 0.04$. The learning rate schedule follows a cosine decay with linear warm-up over 100,000 steps, reaching a peak learning rate of $2 \times 10^{-4}$ before decaying to zero.

## 3.3 THE WAVJEPA-NAT FRAMEWORK

The proposed WavJEPA-Nat is a multi-channel extension of WavJEPA (illustrated in Appendix D). While the overall approach is similar, WavJEPA-Nat is equipped with two waveform encoders and utilizes a 2D instead of a 1D positional embedding to ensure capturing both intra- and inter-channel information. As before, WavJEPA-Nat's objective is to predict the latent representation of target blocks from latent representation of the context block. Crucially, for WavJEPA-Nat, target blocks and the context block indices are shared across *both* channels of the embedded waveform $w$.

**Data and sound wave embeddings:** We use the pipeline of Yuksel et al. (2025) to transform AudioSet sound clips into naturalistic, spatialized sound scenes with reverberation and noise. In brief, we simulate naturalistic, spatialized scenes by using the room impulse response (RIR) simulator and binaural renderer provided by Soundspaces 2.0 (Chen et al., 2022), resulting in two-channel sound clips containing naturalistic spatial cues. To each sound scene, we add similarly spatialized noise clips from the WHAMR! database (Maciejewski et al., 2020). A full description of the sound scene generation can be found in Appendix D.

Each two-channel sound wave $x(t) \in \mathbb{R}^{T \times 2}$ corresponding to a naturalistic scene is transformed by two independent waveform encoders into embeddings $w_1 \in \mathbb{R}^{N \times 768}$ and $w_2 \in \mathbb{R}^{N \times 768}$. The hyperparameters of the waveform encoders are identical to those of WavJEPA. The embedded waveforms $w_1$ and $w_2$ are subsequently concatenated to form $w^{2N \times 768}$.

**Learning inter-channel dependencies:** Instead of adding 1D fixed positional embeddings to $w$ as in the original WavJEPA framework, we now add 2D sinusoidal positional embeddings that explicitly encode both inter-channel and intra-channel positional information. The sampling procedure for obtaining a context block and target blocks is similar to WavJEPA, but shared along the channels. This procedure forces WavJEPA-Nat to predict the latent embedding of the same time step for both channels .

**Pre-training:** As for WavJEPA, we also update the target encoder parameters $\Delta$ for WavJEPA-Nat with the exponential moving average (EMA) of context encoder parameters $\theta$ using a similar schedule for $\tau$. Similarly, we used $K = 8$ for the top $K$ averaging. As the dimensions of $w$, $c_w$ and $z_w$ are twice as large for WavJEPA-Nat, we trained the model with a smaller batch size to avoid out-of-memory errors. Specifically, we used an in-batch sampling factor of 8 and a batch size of 16, resulting in an effective batch size of 128. In agreement with WavJEPA, we trained WavJEPA-Nat for 375 K steps on the same $L2$ objective. The optimization hyper-parameters were kept the same as for WavJEPA.

### 3.4 DOWNSTREAM EVALUATION

**Downstream tasks:** We evaluated WavJEPA and WavJEPA-Nat on two large benchmark task suites for the evaluation of general-purpose audio foundation models: HEAR (Turian et al., 2022) and ARCH (La Quatra et al., 2024). We use the same subset of HEAR benchmark tasks as previously used in Yadav et al. (2024) but added DCASE2016 Task 2 (Mesaros et al., 2018b) as a time stamp-based task to evaluate the audio scene analysis capabilities of the models more in-depth. HEAR and ARCH contain a selection of complementary tasks and datasets for acoustic events and scene analysis, speech, and music. For more detailed description of tasks please see Appendix C.

We additionally evaluated models on NatHEAR (Yuksel et al., 2025), a naturalistic version of the HEAR benchmark suite comprising high-quality simulations of real-world sound scenes with reverberation and noise, spatialized in two formats (either binaural and ambisonics). To accommodate the input format of single-channel models, we utilized the first channel (that is, the omndirectional microphone) of NatHEAR in the Ambisonics format (Zotter & Frank, 2019). For the dual waveform encoder approach of WavJEPANat, we used both channels of NatHEAR in a binaural format.

**Model fine-tuning for downstream evaluation:** For the downstream evaluation on HEAR and ARCH benchmark tasks, we trained a shallow downstream classifier on representations that were extracted after self-supervised pre-training, following the exact fine-tuning procedures detailed by HEAR (Turian et al., 2022) and ARCH (La Quatra et al., 2024). Model weights were frozen after pre-training. Note that the difference between the fine-tuning approaches in HEAR and ARCH causes the differences in performance for tasks that are in both suites, for example, ESC50. Further, to evaluate WavJEPANat on HEAR, we duplicated the single-channel audio recordings of the original HEAR to make the input compatible with the dual waveform encoder architecture of WavJEPANat.

**Down stream evaluation metric $s(m)$:** As the tasks in HEAR and ARCH vary considerably in terms of evaluation criteria and difficulty level, we calculate for each model $m$ a generalizability metric $s(m)$ to give an impression of the overall performance of a model, similar to Yang et al. (2021). This metric effectively ranks models as a function of the maximum improvement they obtain over the baseline model, normalized by the difference in scores between SOTA and the baseline for the specific task (see Appendix B). The baseline used here is HEAR-Naive, consisting of mel-spectrogram representations. For calculating this score, we included all the evaluated methods in all upcoming sections, including ablations.

**Model comparison:** We compare the performance of WavJEPA to state-of-the-art self-supervised models using transformer architectures for representation learning from raw waveforms. We include Wav2Vec2.0 (Baevski et al., 2020), HuBERT (Hsu et al., 2021), WavLM (Chen et al., 2021), Data2Vec (Baevski et al., 2022), all pre-trained on large quantities of speech data. We furthermore include the recently released versions of Wav2Vec2.0 and HuBERT pre-trained on AudioSet (La Quatra et al., 2024) to assess their ability to learn general-purpose audio representations. For all models, we include both the *Base* (approximately 90 m parameters) and the *Large* version (300 m parameters). In comparison, WavJEPA has 90 m parameters.

### 3.5 ABLATIONS:

To identify the critical parameters for a successful learning of general-purpose audio representations with the WavJEPA framework, we conducted comprehensive ablation studies on the pre-training parameters. Specifically, we examined the effect of sampling parameters for target and context blocks ($p_{target}$, $M_{context}$ and $M_{target}$) and the effectiveness of top-$K$ layer averaging for training targets.

For WavJEPANat, we systematically assessed the impact of the ratio between clean and natural-istic sound scenes in the pre-training data. For all ablation studies, pre-training and downstream evaluation settings were similar to those of WavJEPA and WavJEPANat.

## 4 RESULTS

### 4.1 PERFORMANCE ON DOWNSTREAM TASKS

As shown in Table 1 and Table 2, WavJEPA surpasses all state-of-the-art models on HEAR ($s(m)$ = 66.0) and ARCH ($s(m)$ = 92.3). Base models pre-trained on speech score low on both HEAR and ARCH, but improve slightly when pre-trained on AudioSet. This demonstrates that, besides a lack of generalization to out-of-distribution downstream tasks when pre-trained on speech data, these models fail to learn robust general-audio representations from AudioSet pre-training. Among the Large models, WavLM generalizes best to HEAR. It is conceivable that this is a consequence of the size and diversity of the large-scale speech dataset that WavLM *Large* was pre-trained on Chen et al. (2021). HuBERT Large obtained the best score on ARCH when pre-trained on AudioSet.

Table 1: Performance on HEAR benchmark suite. Values represent either the primary score (in case no cross-validation scheme was specified) or the mean ± standard deviation calculated with the *k*-fold cross-validation scheme specified by HEAR. For each task, the best performance per pre-training dataset is highlighted in bold. The best overall performance for a given task (i.e., across pre-training datasets) is highlighted with a light-blue background. *Base* and *Large* refers to the total model parameters, $\sim 90\,$m and $\sim 300\,$m respectively.

| Model | Size | Acoustic Events and Scene Analysis | | | | Speech | | | NS | Music | | | s(m) |
| | | DCASE | FSD50K | LC | ESC-50 | CD | VL | SC-5 | | BO | Mri-S | Mri-T | |
|---|---|---|---|---|---|---|---|---|---|---|---|---|---|
| **Baseline** | | | | | | | | | | | | | |
| HEAR-Naive | N/A | 7.6 | 12.5 | 40.3 ± 1.2 | 27.4 ± 3.3 | 36.7 ± 2.5 | 16.0 ± 3.4 | 13.3 | 89.2 | 97.1 ± 3.2 | 94.2 ± 1.1 | 93.7 ±0.3 | 0.0 |
| **Speech pre-training** | | | | | | | | | | | | | |
| Wav2Vec2.0 | B | 23.5 | 29.4 | 69.9 ± 2.1 | 46.4 ± 1.8 | 57.3 ± 1.1 | 34.9 ± 2.4 | 85.3 | 17.4 | 81.4 ± 4.8 | 90.7 ± 0.8 | 77.0 ± 0.9 | 30.9 |
| HuBERT | B | 78.0 | 32.8 | 63.3 ± 1.2 | 58.6 ± 2.8 | 71.2 ± 1.2 | 65.2 ± 2.9 | 94.0 | 19.8 | 93.2 ± 5.9 | 94.6 ± 0.4 | 85.0 ± 2.5 | 47.3 |
| WavLM | B | 27.0 | 25.7 | 61.3 ± 2.3 | 49.5 ± 3.8 | 64.3 ± 1.3 | 60.1 ± 3.2 | 93.6 | 16.0 | 84.3 ± 6.3 | 88.8 ± 1.0 | 76.8 ± 0.5 | 35.1 |
| Data2Vec | B | 46.5 | 15.2 | 47.9 ± 1.2 | 28.0 ± 2.8 | 55.7 ± 1.0 | 44.9 ± 3.1 | 88.5 | 14.0 | 78.4 ± 4.1 | 85.1 ± 0.7 | 70.5 ± 3.3 | 23.6 |
| Wav2Vec2.0 | L | 66.0 | 34.8 | 64.6 ± 1.9 | 59.8 ± 1.5 | 65.7 ± 0.8 | 53.3 ± 6.3 | 75.8 | 40.6 | 93.6 ± 2.6 | 94.8 ± 0.5 | 82.4 ± 3.0 | 42.5 |
| HuBERT | L | 34.8 | 31.4 | 63.8 ± 1.3 | 60.4 ± 3.0 | 71.0 ± 1.2 | 69.0 ± 2.8 | 84.8 | 20.4 | 93.6 ± 3.0 | 95.3 ± 0.8 | 82.5 ± 2.0 | 44.3 |
| WavLM | L | 77.4 | 40.1 | 69.4 ± 2.1 | 66.6 ± 2.5 | 76.3 ±2.2 | 79.2 ±3.9 | 93.8 | 18.2 | 93.6 ± 5.4 | 95.8 ± 0.8 | 90.1 ± 1.0 | 58.1 |
| Data2Vec | L | 40.8 | 18.7 | 50.9 ± 1.7 | 34.4 ± 2.5 | 62.8 ± 1.6 | 60.0 ± 4.9 | 86.1 | 14.4 | 80.1 ± 8.5 | 84.7 ± 2.6 | 65.6 ± 3.1 | 29.0 |
| **AudioSet pre-training** | | | | | | | | | | | | | |
| Wav2Vec2.0 | B | 52.0 | 34.7 | 60.4 ± 1.7 | 58.9 ± 1.9 | 56.3 ± 1.3 | 27.9 ± 4.6 | 72.1 | 42.0 | 86.0 ± 9.6 | 92.9 ± 1.4 | 77.3 ± 0.5 | 31.9 |
| HuBERT | B | 86.2 | 41.1 | 63.5 ± 3.4 | 69.1 ± 1.6 | 69.5 ± 1.2 | 53.3 ± 3.1 | 83.5 | 38.8 | 91.5 ± 8.8 | 95.6 ± 0.5 | 90.4 ± 0.8 | 51.1 |
| Wav2Vec2.0 | L | 82.6 | 47.8 | 73.6 ± 1.2 | 72.6 ± 2.1 | 68.2 ± 1.7 | 42.2 ± 6.0 | 83.9 | 30.8 | 91.5 ± 5.0 | 96.5 ± 0.3 | 88.7 ± 2.5 | 55.9 |
| HuBERT | L | 86.2 | 45.4 | 75.2 ± 1.4 | 66.3 ± 4.6 | 70.1 ± 0.8 | 39.6 ± 3.6 | 85.7 | 38.6 | 91.6 ±9.6 | 97.3 ±0.5 | 89.6 ± 2.3 | 57.7 |
| WavJEPA | B | 93.9 | 54.4 | 76.7 ±2.4 | 86.5 ± 3.3 | 71.0 ± 0.8 | 49.8 ± 3.4 | 90.0 | 34.4 | 89.4 ± 5.4 | 97.3 ±0.4 | 88.5 ± 0.5 | 66.0 |

Table 2: Performance on ARCH benchmark suite. Values and colors as in Table 1.

| Model | Size | Acoustic Events and Scene Analysis | | | | Music | | | | Speech | | | | s(m) |
| | | ESC-50 | US8K | FSD50K | VIVAE | FMA | MTT | IRMAS | MS-DB | RAVDESS | AM | SLURP | EMOVO | |
|---|---|---|---|---|---|---|---|---|---|---|---|---|---|---|
| **Baseline** | | | | | | | | | | | | | | |
| HEAR-Naive | N/A | 13.0 | 36.0 | 2.2 | 22.0 | 39.0 | 9.9 | 19.9 | 35.2 | 22.6 | 45.7 | 5.4 | 18.4 | 0.0 |
| **Speech pre-training** | | | | | | | | | | | | | | |
| Wav2Vec2.0 | B | 45.7 | 55.5 | 19.4 | 31.5 | 50.5 | 37.6 | 35.1 | 66.1 | 55.3 | 86.4 | 14.4 | 31.8 | 49.7 |
| WavLM | B | 49.9 | 61.8 | 17.6 | 36.3 | 48.7 | 34.9 | 32.6 | 54.2 | 67.9 | 99.5 | 31.0 | 43.1 | 68.0 |
| HuBERT | B | 58.9 | 67.3 | 24.5 | 40.5 | 54.6 | 38.8 | 36.7 | 65.3 | 65.3 | 99.6 | 33.8 | 40.5 | 59.7 |
| Data2Vec | B | 23.6 | 45.6 | 10.1 | 30.2 | 40.6 | 27.6 | 25.9 | 50.7 | 48.0 | 99.1 | 43.6 | 27.3 | 38.8 |
| Wav2Vec2.0 | L | 13.1 | 42.7 | 5.8 | 22.0 | 41.7 | 21.0 | 19.9 | 50.2 | 11.6 | 45.7 | 7.3 | 19.3 | 8.6 |
| WavLM | L | 67.2 | 70.9 | 32.2 | 42.5 | 61.1 | 41.3 | 42.5 | 68.0 | 71.8 | 99.8 | 45.3 | 45.3 | 75.8 |
| HuBERT | L | 64.0 | 70.0 | 29.5 | 41.0 | 54.8 | 38.4 | 36.8 | 64.1 | 72.6 | 99.9 | 45.3 | 43.8 | 81.5 |
| Data2Vec | L | 25.4 | 49.2 | 10.8 | 30.6 | 43.5 | 28.5 | 27.1 | 44.2 | 45.1 | 99.2 | 28.6 | 23.1 | 35.1 |
| **AudioSet pre-training** | | | | | | | | | | | | | | |
| W2V2 | B | 52.6 | 70.5 | 21.3 | 31.3 | 59.5 | 37.9 | 35.9 | 64.6 | 45.9 | 88.1 | 11.0 | 30.8 | 53.8 |
| HuBERT | B | 68.8 | 79.1 | 31.1 | 40.1 | 65.9 | 43.4 | 47.7 | 67.8 | 63.5 | 98.8 | 20.5 | 33.4 | 75.5 |
| Wav2Vec 2.0 | L | 74.4 | 79.0 | 37.6 | 39.7 | 66.6 | 44.5 | 49.9 | 76.9 | 59.5 | 99.4 | 17.7 | 38.2 | 80.0 |
| HuBERT | L | 71.5 | 75.6 | 37.4 | 44.3 | 67.5 | 43.4 | 50.5 | 77.8 | 73.3 | 99.6 | 20.5 | 38.6 | 83.9 |
| WavJEPA | B | 83.9 | 83.5 | 48.0 | 44.06 | 68.2 | 46.0 | 59.0 | 79.5 | 62.5 | 99.5 | 23.3 | 46.6 | 92.3 |

**Audio scene analysis and acoustic events:** Inspecting performances at the task level demonstrates that WavJEPA performs exceptionally well on acoustic events and audio scene analysis. On tasks

such as sound event detection (DCASE 2016 Task 2), WavJEPA improves the SOTA by 8.9 %, and on audio event multi-labeling task FSD50K - a very challenging task - WavJEPA increases the SOTA by 13.8 %. For environmental sound classification, WavJEPA's accuracy is 19.1 % higher than the next best performing model (WavLM Large).

**Speech:** The tasks covered by the pre-training data has, as expected, a large impact on the speech-related downstream tasks. In particular, WavLM Large pre-trained on speech data obtains the highest performance on HEAR speech tasks, while HuBERT Large scores best on ARCH speech tasks (followed by WavLM Large). However, among the Base models pre-trained on AudioSet, WavJEPA performs best on several of the HEAR speech tasks, including spoken command classification (SC5) and emotion recognition (CD), as well as on most of the ARCH speech tasks, including spoken digit recognition (AudioMNIST, AM), intent classification (SLURP) and emotion recognition (EMOVO). Moreover, WavJEPA outperforms several Base models pre-trained on speech, both on HEAR and on ARCH speech tasks, illustrating the generalization of WavJEPA to speech data.

**Music:** WavJEPA obtains the highest performance on all music tasks in the ARCH benchmark suite. However, we find that models pre-trained on AudioSet do not unequivocally perform better on HEAR music tasks as well. This may be related to the type of music tasks. That is, while ARCH includes music tasks of a general nature (genre classification, tagging and instrument recognition (La Quatra et al., 2024)), HEAR includes niche music tasks including pitch classification and percussion classification. These types of tasks appear less suitable for WavJEPA representations, as WavJEPA obtains SOTA performance on just one of the HEAR music tasks.

**Model efficiency:** Crucially, Figure 2 demonstrates that WavJEPA requires only a fraction of the pre-training data to surpass other time-domain models on HEAR and ARCH, despite the small model size of only 90 m parameters. Furthermore, we find that WavJEPA's performance scales with the amount of pre-training data (Figure 2).

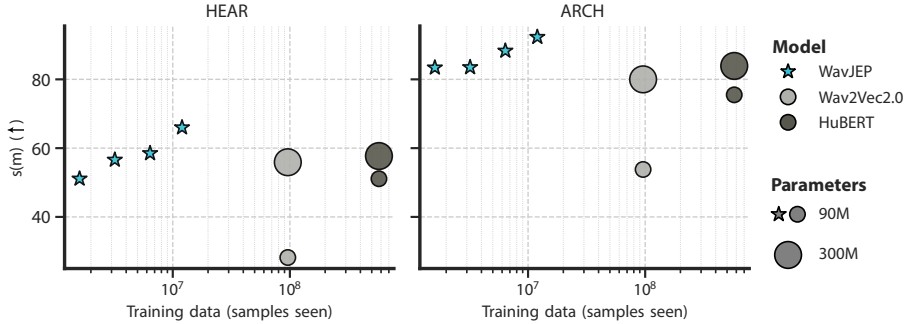

Figure 2: **Downstream task performance** $s(m)$ **vs. pre-training data (AudioSet).** Symbols depict performance $s$ for HEAR (left panel) and for ARCH (right panel) as a function of number of samples seen during pre-training. Symbol size reflects the number of model parameters. For WavJEPA, we depict performance after 50 k, 100 k, 200 k and 375 k training steps.

## 4.2 EVALUATION ON NATURALISTIC SCENES

**Transferability to naturalistic scenes:** Table 3 shows that the performance of all models is lower in naturalistic scenes. However, we find that, even when trained on non-naturalistic data, WavJEPA generalizes best to naturalistic scenes ($s = 62.1$) and performs almost similarly on NatHEAR as on HEAR ($\Delta s = -3.9$). This demonstrates that the high-level semantic representation learning approach of the JEPA architecture can successfully learn robust representations which generalize to noisy and reverberant environments. Further, WavJEPA excels specifically on tasks related to audio scene analysis and acoustic events on Nat-HEAR Table 3. WavJEPA also surpasses other Base and Large models trained on AudioSet on most speech and music tasks in NatHEAR, but not the WavLM Large model on Nat-HEAR speech tasks.

Table 3: Generalization to naturalistic scenes (NatHEAR benchmark suite). Values and colors as in Table 1.

| Model | Size | Acoustic Events and Scene Analysis | | | | Speech | | | Music | | | | s(m) |
| | | DCASE | FSD50K | LC | ESC-50 | CD | VL | SC-5 | NS | BO | Mri-S | Mri-T | |
|---|---|---|---|---|---|---|---|---|---|---|---|---|---|
| **Baseline** | | | | | | | | | | | | | |
| HEAR-Naive | N/A | 0.7 | 8.7 | 26.9 ± 1.9 | 16.1 ± 2.0 | 28.8 ± 2.6 | 12.7 ± 3.6 | 12.3 | 78.6 | 88.6 ±6.0 | 80.5 ± 0.7 | 75.0 ±4.0 | 0.0 |
| **Speech pre-training** | | | | | | | | | | | | | |
| W2V2 | B | 32.0 | 23.0 | 54.6 ± 1.9 | 36.4 ± 2.9 | 48.6 ± 0.6 | 27.2 ± 1.6 | 78.9 | 15.2 | 71.2 ± 6.4 | 75.7 ± 0.5 | 45.9 ± 0.6 | 32.7 |
| HuBERT | B | 57.6 | 26.6 | 52.5 ± 2.2 | 49.5 ± 2.2 | 57.4 ± 1.1 | 46.8 ± 3.4 | 89.2 | 16.0 | 77.1 ± 6.0 | 78.2 ± 0.7 | 52.4 ± 1.6 | 44.6 |
| WavLM | B | 25.3 | 20.5 | 52.1 ± 0.6 | 41.4 ± 2.1 | 52.3 ± 1.5 | 47.9 ± 4.6 | 89.9 | 11.2 | 61.4 ± 7.2 | 69.3 ± 0.9 | 39.0 ± 2.0 | 37.3 |
| D2V | B | 15.5 | 12.0 | 39.1 ± 1.1 | 19.1 ± 1.5 | 42.8 ± 0.9 | 30.5 ± 1.5 | 71.9 | 4.6 | 58.5 ± 3.2 | 55.5 ± 1.7 | 36.1 ± 1.2 | 19.7 |
| W2V2 | L | 52.7 | 26.6 | 53.0 ± 0.9 | 42.5 ± 3.5 | 50.9 ± 1.0 | 33.2 ± 5.0 | 58.7 | 30.6 | 69.5 ± 5.7 | 77.4 ± 0.8 | 54.8 ± 2.7 | 35.6 |
| HuBERT | L | 16.7 | 23.4 | 52.3 ± 0.3 | 48.7 ± 0.7 | 50.5 ± 1.2 | 42.9 ± 3.9 | 69.9 | 14.6 | 75.0 ± 5.7 | 84.4 ± 1.4 | 54.8 ± 1.4 | 38.6 |
| WavLM | L | 75.6 | 34.1 | 58.7 ± 1.0 | 56.5 ± 2.8 | 63.7 ± 1.6 | 64.5 ± 2.7 | 92.6 | 14.6 | 76.6 ± 7.6 | 82.7 ± 0.6 | 54.9 ± 1.4 | 58.5 |
| D2V | L | 40.6 | 15.0 | 43.5 ± 0.5 | 22.9 ± 2.8 | 53.7 ± 1.5 | 43.1 ± 4.6 | 73.5 | 10.4 | 63.1 ± 6.6 | 59.0 ± 5.2 | 33.2 ± 3.1 | 30.1 |
| **AudioSet pre-training** | | | | | | | | | | | | | |
| W2V2 | B | 33.1 | 27.7 | 51.0 ± 1.2 | 48.1 ± 2.1 | 43.9 ± 2.2 | 22.3 ± 1.5 | 60.1 | 21.2 | 75.8 ± 6.0 | 74.4 ± 1.6 | 45.2 ± 1.5 | 30.5 |
| HuBERT | B | 69.8 | 34.7 | 53.0 ± 1.0 | 56.6 ± 2.5 | 48.9 ± 1.6 | 40.6 ± 2.0 | 76.3 | 29.8 | 80.1 ± 5.8 | 79.3 ± 1.1 | 52.8 ± 1.2 | 44.3 |
| W2V2 | L | 65.2 | 39.8 | 57.6 ± 1.5 | 56.1 ± 2.4 | 52.4 ± 1.0 | 26.2 ± 5.1 | 74.2 | 17.8 | 74.1 ± 6.2 | 81.3 ± 0.9 | 52.5 ± 2.5 | 45.2 |
| HuBERT | L | 68.1 | 37.8 | 58.1 ± 1.9 | 55.3 ± 4.1 | 54.1 ± 0.5 | 29.5 ± 2.6 | 77.6 | 26.2 | 77.9 ± 7.2 | 87.2 ± 1.2 | 59.9 ± 2.0 | 52.4 |
| WavJEPA | B | 83.1 | 47.0 | 59.7 ±1.8 | 76.0 ± 2.8 | 57.6 ± 0.4 | 35.0 ± 3.0 | 82.2 | 25.0 | 82.2 ± 4.4 | 87.1 ± 0.7 | 57.0 ± 1.2 | 62.1 |

**Impact of pre-training on naturalistic scenes:**  We find that pre-training on naturalistic scenes improves the downstream performance on HEAR as well as NatHear. In particular, Table 3 shows that WavJEPANat performs better than WavJEPA on both HEAR and NatHEAR on almost all tasks. Moreover, WavJEPANat exhibits superior performance compared to all other models on both HEAR ($s = 60.0$, compare to Table 1) and NatHEAR ($s = 61.2$, compare to Table 3), even though pre-trained with only half the batch size. This suggests that WavJEPANat could benefit from further upscaling.

Table 4: Impact of naturalistic pre-training on HEAR and NatHEAR performance. Note that WavJEPANat was pre-trained with a lower batch size than the original WavJEPA. For comparison, we depict the results of WavJEPA pre-trained with a similar batch size as WavJEPANat subsection 3.4. We indicate the best performing model per benchmark in **bold**.

| Model | Size | Acoustic Events and Scene Analysis | | | | Speech | | | Music | | | | s(m) |
| | | DCASE | FSD50K | LC | ESC-50 | CD | VL | SC-5 | NS | BO | Mri-S | Mri-T | |
|---|---|---|---|---|---|---|---|---|---|---|---|---|---|
| **Performance on HEAR** | | | | | | | | | | | | | |
| WavJEPA | B | **92.3** | **51.2** | 69.5 ±2.4 | 78.7 ±2.7 | 64.5 ±1.3 | **43.5** ±3.0 | **89.2** | 25.8 | 89.8 ±6.6 | 96.8 ±0.4 | 86.2 ±0.5 | 58.3 |
| WavJEPA-Nat | B | 91.6 | 48.7 | **72.4** ±1.8 | **80.2** ±1.7 | **65.9** ±0.7 | 39.7 ±2.4 | 87.4 | **33.4** | **96.2** ±5.3 | **97.4** ±0.5 | **90.4** ±0.8 | **60.0** |
| **Performance on Nat-HEAR** | | | | | | | | | | | | | |
| WavJEPA | B | 80.6 | **43.0** | 56.1 ± 2.9 | 68.4 ± 3.1 | 52.2 ± 1.8 | **28.5** ± 2.6 | 81.5 | 17.0 | 79.6 ± 6.2 | 86.9 ± 0.8 | 58.2 ± 1.0 | 55.8 |
| WavJEPANat | B | **86.0** | 42.4 | **59.2** ± 1.6 | **72.6** ± 2.5 | **56.3** ± 1.2 | 27.9 ± 3.3 | **81.9** | **26.8** | **87.7** ± 3.6 | **89.3** ± 0.4 | **63.5** ± 0.9 | **61.2** |

## 4.3 ABLATION STUDIES

**Ratio of clean versus naturalistic pre-training data:**  Prior work on spectrogram-based representation learning showed that downstream task performance in scenes with reverberation benefits from pre-training on a mix of naturalistic, reverberant sounds and clean sounds in comparison to pre-training exclusively on naturalistic, reverberant scenes (Devnani et al., 2024). We investigated to what extent pre-training on a mixture of clean and naturalistic sound scenes affected the performance of WavJEPANat on HEAR and NatHEAR. Figure 3 (left panel) shows that the higher the ratio of clean data ($\lambda$), the lower the performance of WavJEPANat on both HEAR and Nat-HEAR. This demonstrates that WavJEPANat learns more robust and generalizable representations from naturalistic scenes and, importantly, that pre-training on naturalistic scenes boosts performance on downstream tasks comprising only clean sounds as well. These results demonstrate that combining the high-level semantic representation learning of the JEPA architecture with a dual waveform encoder as in WavJEPANat can learn robust audio representations from noisy and reverberant data, enhancing performance on both clean sounds as well as noisy and reverberant scenes.

**Top-K averaging:** We assessed whether averaging training targets over the top-$K$ layers improved the quality and robustness of WavJEPA's learned representations for $K = 1$, 4, 8, and 12 (i.e., all layers) (Baevski et al., 2020). The results show that top-$K$ averaging indeed improves downstream performance on all HEAR tasks, although the range of improvement varied across tasks, see

Figure 3 (middle panel). Moreover, for some scene analysis and speech tasks (LibriCount, ESC50, and Crema-D), performance peaked at $K = 8$ and decreased again for $K = 12$, while other tasks did not exhibit a difference in performance between $K = 8$ and $K = 12$. These findings indicate that top-$K$ layer averaging substantially improves downstream performance, but that an optimal value of $K$ is task-dependent.

**Target length and context length:** The length of segments sampled for the training targets ($M_{target}$) and segments sampled for the context block ($M_{context}$) impacts their degree of distribution. A small value of $M$ leads to a more distributed context block or training target, while a large $M$ results in a less distributed context block or training target. We found that $M_{context}$ had little impact on the downstream task performance ($s(m) = [66.2, 66.0, 64.0]$ for $M = [5, 10, 15]$, see Appendix E). In contrast, we found that highly distributed training targets were consistently suboptimal for scene analysis and speech tasks, see Figure 3 (right panel).

**Target probability:** A higher sampling probability for target indices ($p_{target}$) results in larger training targets and a smaller context block (as the proportion of $w$ sampled as target indices goes up, while the proportion of $w$ sampled as context indices goes down, see Appendix F). Ablating $p_{target}$ revealed some variation in downstream performance, although not substantially: $p_{target} = [0.15, 0.20, 0.25, 0.30]$ resulted in $s(m) = [64.8, 65.9, 66.0, 63.0]$, see Appendix E. These findings suggest that sampling target indices with a probability between 0.15 and 0.25 is optimal, whereas a higher sampling probability reduces WavJEPA's representation learning capacity.

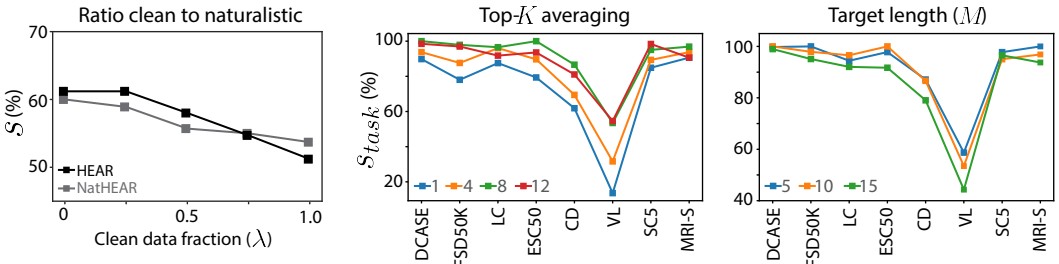

Figure 3: **Ablation studies.** The left panel compares the performances on HEAR and NatHEAR for the WavJEPANat architecture as a function of the ratio ($\lambda$) between clean and naturalistic scenes in the pre-training data. The middle panel depicts the impact of the top-$K$ averaging parameter per HEAR task for WavJEPA. The right panel compares the impact of target length ($M_{target}$) per task. The middle and right panels include only HEAR tasks for which WavJEPA performed better than baseline for ease of visualization.

## 5 DISCUSSION AND CONCLUSION

We presented WavJEPA, a state-of-the-art audio foundation model that leverages self-supervised semantic learning to obtain robust general-purpose audio representations from raw waveforms. WavJEPA's results highlight the superior performance of semantic audio representation learning in comparison with representation learning at the speech unit or token level, as is common in existing time-domain speech representation learning approaches. Moreover, WavJEPA is highly efficient, requiring only a fraction of the training data in comparison to other time-domain models. Furthermore, our results demonstrate that WavJEPA is robust to noise and reverberation, emphasizing the suitability of semantic learning for deriving representations that generalize across acoustic environments. As WavJEPA's speech representation learning could still be improved in comparison to Large speech models, we plan to investigate the benefit of pretraining WavJEPA on a combination of sound databases such as AudioSet and speech databases. Taken together, WavJEPA unlocks general-purpose audio representation learning in the time domain, opening up avenues towards real-time audio foundation models. and high-quality audio generation audio foundation models. WavJEPA also highlights the potential of time-domain audio foundation models for high-quality speech stream generation in speech separation and speech denoising applications, as well other generative audio tasks.

## 6 REPRODUCIBILITY STATEMENT

All the code, datasets, and checkpoints for WavJEPA, WavJEPA-Nat, and for ablation studies will be made publicly available on `https://TBD.com` on an open source license. Furthermore, we plan on hosting WavJEPA models on HuggingFace `https://huggingface.co/` for ease of use and reproducibility.

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

# APPENDIX

## A DETAILED TRAINING SPECIFICATIONS

Table 5: **Pre-training specifications**

| Configuration | Pre-training |
|---|---|
| Optimizer | AdamW |
| Optimizer momentum | $\beta_1 = 0.9, \beta_2 = 0.98$ |
| Weight decay | 0.04 |
| Base learning rate | 0.0004 |
| Learning rate schedule | linear-warmup + cosine decay |
| Minimum learning rate | 0.0 |
| Dropout | 0. |
| Warm-up steps | 100,000 |
| Total steps | 375,000 |
| Early Stopping | N/A |
| Batch size | 32 |
| Accelerators | 2 x GPU H100 92 GB |
| Target-Encoder & Context Encoder | ViT-B |
| Predictor | ViT-S |
| Target-Encoder & Context-Encoder Parameters | 86 M |
| Predictor Parameters | 22 M |
| Waveform Encoder | Convolutions with 512 channels, strides (5,2,2,2,2,2) and kernel widths (10,3,3,3,3,2) |
| Waveform Encoder Parameters | 4 M |

## B DOWNSTREAM EVALUATION METRIC

Similar to the procedure in SUPERB (Yang et al., 2021), let $s_t$ be the metric for task $t$. We then calculate the generalizability metric HEAR $s(m)$, ARCH $s(m)$ and Nat-HEAR $s(m)$ for model $m$ as:

$$s(m) = \frac{100}{T} \sum_{t}^{T} \frac{s_t(m) - s_t(baseline)}{s_t(SOTA) - s_t(baseline)}$$

Intuitively, this metric ranks the improvement of models over the baseline as a function of the maximum improvement over the baseline obtained by the current state-of-the-art. Note that we replace $s_t(m)$ for task $t$ of model $m$ with 0 when the model scores below baseline performance for task $t$. Similarly, when $s_t(SOTA)$ is lower than baseline for task $t$, we set for all models $s_t$ for this task to 0. In this way, all values are restricted to a range of improvement between 0 % and 100 %.

## C HEAR, NAT-HEAR AND ARCH TASKS

Table 6 illustrates the abbreviations, task description, and the type that we have utilized to benchmark our models. Furthermore, Table 2 demonstrates the specification of ARCH tasks.

Table 6: Overview of the HEAR and Nat-HEAR tasks.

| Abbreviation | Task Name | Description | Type |
|---|---|---|---|
| DCASE | DCASE-2016 Task 2 (Mesaros et al., 2018a) | Event detection of overlapping office sounds in synthetic mixtures | Scene Analysis |
| FS50K | FSD50k (Fonseca et al., 2022) | Multilabel, large scale audio tagging | Environmental Sound Classification |
| LC | LibriCount (Stöter et al., 2018) | Speaker Count Identification, Simulated Cocktail Party | Scene Analysis |
| ESC-50 | ESC-50 (Piczak) | Environmental Sound Classification | Environmental Sound Classification |
| CD | Crema-D (Cao et al., 2014) | Emotion Recognition | Speech Analysis |
| VL | VoxLingua107 Top10 (Valk & Alumäe, 2021) | Spoken language identification | Speech Analysis |
| SC-5 | Speech Command 5h (Warden, 2018) | Keyword Spotting, reduced training subset | Speech Analysis |
| NS | NSynth Pitch 5h (Engel et al., 2017) | Pitch Classification, reduced training subset | Music |
| BO | Beijing Opera (Tian et al., 2014) | Classifying percussion instruments | Music |
| Mri-S | Mridangam Stroke (Anantapadmanabhan et al., 2013) | Stroke classification in pitched percussion instruments | Music |
| Mri-T | Mridangam Tonic (Anantapadmanabhan et al., 2013) | Tonic classification in pitched percussion instruments | Music |

Table 7: Datasets included in ARCH with their corresponding domain, classification task types (single S or multi-label M), number of samples, average duration, and number of classes.

| Dataset | Domain | Task | Samples | Avg duration | Classes |
|---|---|---|---|---|---|
| ESC-50 (Piczak) | Environmental Sound Classification | S | 2000 | 5.0 s | 50 |
| US8K (Salamon et al., 2014) | Environmental Sound Classification | S | 8732 | 3.61 s | 10 |
| FSD50K (Fonseca et al., 2022) | Environmental Sound Classification | M | 51197 | 7.64 s | 200 |
| VIVAE (Holz et al., 2022) | Environmental Sound Classification | S | 1085 | 0.90 s | 6 |
| FMA (Defferrard et al., 2017) | Music | S | 8000 | 29.98 s | 8 |
| MTT (Law et al., 2009) | Music | M | 21108 | 29.12 s | 50 |
| IRMAS (Bosch et al., 2012) | Music | M | 8278 | 5.73 s | 11 |
| MS-DB (Lostanlen & Cella, 2016) | Music | S | 21571 | 2.97 s | 8 |
| RAVDESS (Livingstone & Russo, 2018) | Speech Analysis | S | 1440 | 3.70 s | 8 |
| AM (Becker et al., 2024) | Speech Analysis | S | 30000 | 0.64 s | 10 |
| SLURP (Bastianelli et al., 2020) | Speech Analysis | S | 72396 | 2.85 s | 77 |
| EMOVO (Costantini et al., 2014) | Emotion Recognition | S | 588 | 3.12 s | 7 |

## D WAVJEPA-NAT FRAMEWORK

To train WavJEPA-Nat on naturalistic scenes, we make use of the natural scenes introduced by (Yuksel et al., 2025). In particular, (Yuksel et al., 2025) provide a set of 85,000 binaural room impulse responses (BRIRs) for rendering two-channel sound scenes consisting of a sound source sampled from AudioSet and a noise source from WHAMR! (either localized or diffuse). A brief description of BRIRs and naturalistic sound scenes is provided here, a full description can be found in the original paper.

The BRIRs encompass 85 houses from MatterPort3D (Chang et al., 2017). Room Impulse Responses (RIRs) are simulated for the different rooms in the houses with the Monte Carlo ray tracing simulator of SoundSpaces2.0 (Chen et al., 2020; 2022). Naturalistic scenes are generated by randomly positioning a listener, sound source and noise source in a room (1,000 for each house). Noise sources were either added as localized or as a diffuse noise field. The SoundSpaces2.0 simulator combined the simulated RIRs for each scene with a head-related imulse response (HRIR) to render a binaural RIR (BRIR). The BRIR captures the characteristics of both the room acoustics and binaural hearing. In total, the set consists of 85,000 BRIRs corresponding to 85,000 naturalistic sound scenes with $RT_{60}$ ( reverberation strength) ranging between 0.2 and 0.5.

**Simulating naturalistic sound scenes:** We used the naturalistic sound scene generation pipeline introduced by Yuksel et al. (2025). A brief description of the pipeline is included here, a full description can be found in the original paper.

The pipeline makes use of the high-resolution 3D meshes of 85 houses from MatterPort3D [REF] to simulate room impulse responses (RIRs) for many different rooms with the Monte Carlo ray tracing simulator of SoundSpaces2.0 [REF]. A naturalistic scene (1,000 for each house) is subsequently generated by randomly positioning a listener, sound source and noise source in a room. Noise sources were either added as localized or as a diffuse noise field. The SoundSpaces2.0 simulator combines the simulated RIRs for each scene with a head-related imulse response (HRIR) to render a binaural RIR (BRIR). The BRIR captures the characteristics of both the room acoustics and binaural hearing. In this way, we generated Here, we used the state-of-art Monte Carlo ray tracing RIR simulator provided by SoundSpaces to simulate RIRs for a wide variety of rooms. We extracted high-resolution, detailed 3D meshes of houses with various architectural characteristics from Matterport3D as input for the SoundSpaces2.0 simulator. SoundSpaces combines the simulated RIRs with a head-related transfer function (HRTF) to generate a binaural RIR (BRIR), which captures both room specific acoustic properties and binaural hearing properties. Matterport3D contains scans of 90 houses. We discarded five houses for which meshes were not of sufficient quality. For each of the remaining 85 houses, we generated 1,000 naturalistic scenes.

We generated a naturalistic scene by randomly sampling a listener location, a sound source location and a noise source location in the room. Listeners were placed within the room with a randomly sampled head orientation (range [0°, 360°]). We placed the sound source location at a randomly

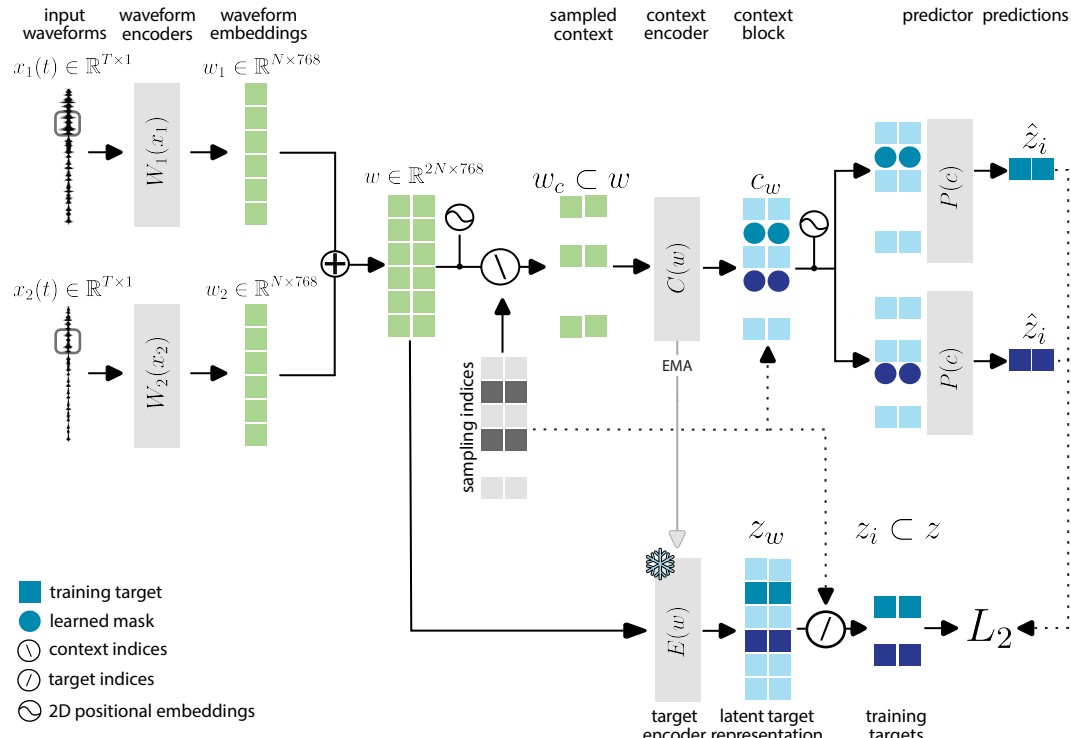

Figure 4: **Robust representation learning from naturalistic sound scenes including noise and reverberation.** WavJEPANat is a multi-channel extension of WavJEPA which uses a dual waveform encoder to learn inter- and intra-channel characteristics and predicts 2D latent target representations from a 2D context block. The weights of the target encoder are not trained but updated using the exponential moving average (EMA) of the weights of the context encoder.

sampled location with respect to the listener (distance range [1.5 m, 5 m]; azimuth range [0°, 360°]; elevation range [-90°, +90°]). Noise could either be localized (50 % of the scenes) or diffuse (50 % of the scenes). For localized noise, we randomly sampled one location in the room. For diffuse noise, we randomly sampled three, four or five locations in the room. We then rendered a set of BRIRs to describe the naturalistic scene. Given sound source location s, listener location r, and receiver head orientation $\theta$, we rendered the BRIR between the listener and the source as BRIR(s, r, $\theta$). Given a number of noise sources $n_i$ with noise source location $\phi_i$ , listener location r, and receiver head orientation $\theta$, we rendered the BRIR between the listener and each noise source as BRIRi($\phi_i$ 140 , r, $\theta$). This procedure resulted in a total of 85,000 sets of BRIRs with $RT_{60}$ ( reverberation strength) ranging between 0.2 and 0.5.

**Training on naturalistic scenes:** Similar to Yuksel et al. (2025), we divided the 85,000 BRIRs for the naturalistic scenes into a train set (70,000 scenes) and a test set (15,000 scenes) for down-stream evaluation (see section experiments). We used the 70,000 naturalistic scenes in the train set to generate a naturalistic version of the unbalanced training set of AudioSet. Specifically, during training we randomly paired every AudioSet clip with a noise sound clip from the WHAMR! background noise database. WHAMR! noise clips longer than 10 s were trimmed to 10 s duration and a linear fade-in/fade-out of 200 ms was added to every WHAMR! noise clip prior to mixing of the sound scene. To create the naturalistic sound scene, we then convolved the sound source BRIR with the AudioSet clip to obtain S, and the noise source BRIR(s) with the WHAM! clip to obtain $N_i$ . In naturalistic scenes with diffuse background noise, the diffuse noise field was generated by summing the noise clips N = P i Ni 183. The naturalistic sound scene S was then calculated as S = T + bN, where b is 184, a scaling parameter introduced to mix target and noise sound clips at a given signal-to-noise ratio of 185 (SNR) ranging between +5 dB and +40 dB.

## E    DETAILED RESULTS ABLATION STUDIES

Table 8: **Ablations for context and training target sampling procedure**. Downstream performance on HEAR benchmark. *Italics* denote modifications with respect to the baseline.

| | WavJEPA | $M_{context}$ | | $M_{target}$ | | $p_{target}$ | | |
|---|---|---|---|---|---|---|---|---|
| $M_{context}$ | 10 | *5* | *15* | 10 | 10 | 10 | 10 | 10 |
| $M_{target}$ | 10 | 10 | 10 | *5* | *15* | 10 | 10 | 10 |
| $p_{target}$ | 0.25 | 0.25 | 0.25 | 0.25 | 0.25 | *0.15* | *0.20* | *0.30* |
| $s(m)$ | 66.0 | 66.2 | 64.0 | 66.9 | 62.9 | 64.8 | 65.9 | 63.0 |

## F    DISTRIBUTION OF TARGET AND CONTEXT SAMPLING

Table 9: **Proportion of sound wave embedding $w$ sampled as context block and as training targets**. Values indicate average and 95 % confidence interval. Note that each sound wave embedding $w$ contains on average 4 training targets.

| $M_{context}$ | $M_{target}$ | $p_{target}$ | Context block indices (%) | Training target indices (%) |
|---|---|---|---|---|
| *Baseline* | | | | |
| 10 | 10 | 0.25 | 19.6 [11.5, 30.0] | 22.7 [17.5, 25.0] |
| *Target Length* | | | | |
| 10 | 5 | 0.25 | 18.8 [11.5, 26.5] | 22.8 [19.5, 25.0] |
| 10 | 15 | 0.25 | 19.9 [11.0, 31.5] | 22.8 [15.5, 30.0] |
| *Context Length* | | | | |
| 5 | 10 | 0.25 | 18.8 [11.5, 27.5] | 22.7 [17.0, 25.0] |
| 15 | 10 | 0.25 | 19.7 [11.0, 30.5] | 22.7 [17.5, 25.0] |
| *Target Probability* | | | | |
| 10 | 10 | 0.15 | 28.1 [18.0, 39.0] | 14.3 [10.5, 15.0] |
| 10 | 10 | 0.20 | 23.2 [13.5, 34.0] | 18.7 [14.0, 20.0] |
| 10 | 10 | 0.30 | 16.7 [10.5, 26.5] | 26.6 [21.0, 30.0] |

