# OpenReview forum: "WavJEPA: Semantic learning unlocks robust audio foundation models for raw waveforms"
_ICLR.cc/2026/Conference — ICLR 2026 Conference Withdrawn Submission_

### Official Review · Reviewer_9LaT · 2025-10-19

**Soundness:** 2
**Presentation:** 2
**Contribution:** 1
**Rating:** 2
**Confidence:** 4

**Summary:**

This work introduces WavJEPA, a waveform-based general-purpose audio representation learning method built upon JEPA, with WavJEPA-nat proposed to specifically tackle real-world acoustic environments.
WavJEPA and WavJEPA-nat are evaluated on HEAR and ARCH benchmarks, showing SOTA performance.

**Strengths:**

1. The idea of WavJEPA generally makes sense, given there's no JEPA variant for waveform.
2. WavJEPA shows SOTA results. The experiments on showing WavJEPA's overall performance are sound.
3. WavJEPA-nat is useful for real-world cases.

**Weaknesses:**

1. Motivation needs to be improved. There have been some mel-spectrum-based JEPA variants [a, b, c], which are very relevant to WavJEPA. Did you get motivated by them? What's your advantage over them? They should be the baselines in the Experiment section, given they also claim superior performance over baseline models adopted by WavJEPA.

2. You mention that the core advantage of WavJEPA is that it "overcomes the limitations of learning representations at the token or speech unit level." (1) What exactly are the "limitations"? I understand there have been some image-based arguments in I-JEPA. However, as your whole work is JEPA on waveform, I would expect in-depth audio-based arguments, paired with qualitative/quantitative analysis on the difference in features learned by WavJEPA and token/unit level methods, instead of only showing that you achieve SOTA performance on benchmarks.

3. The introduction to JEPA in 3rd paragraph of the Related Work Section is a bit superficial. It does not mention JEPA's intuition and designs to achieve the goal. You may also introduce some terminology used by the original paper (e.g., "context" and "target"). Now this paragraph only mentions its good performance ("efficiently learn semantic image representations"). Insufficient introduction to JEPA also undermines the readability of the Methodology section.

4. The novelty (settings, engineering techniques, and insights) of this work seems a bit limited. I would assume engineering techniques (e.g., how to adapt JEPA to the waveform) to be the highlights. While the Introduction, Related Work, and Methodology sections do not show this.

5. Limitation is not discussed.

a. [A-JEPA: Joint-Embedding Predictive Architecture Can Listen](https://arxiv.org/pdf/2311.15830v3)

b. [INVESTIGATING DESIGN CHOICES IN JOINT-EMBEDDING PREDICTIVE ARCHITECTURES FOR GENERAL AUDIO REPRESENTATION LEARNING](https://arxiv.org/pdf/2405.08679)

c. [Audio-JEPA: Joint-Embedding Predictive Architecture for Audio Representation Learning](https://arxiv.org/pdf/2507.02915)

**Questions:**

If authors can successfully address above weaknesses, I am happy to raise my score to 4. It may be hard for me to consider 6 for this paper currently. My main concerns include the lack of motivation, writing (especially in Section 2 & 3, as mentioned in Weakness), and novelty.

---

> ### Author Response · Authors · 2025-12-03
>
> We thank the reviewer for their positive comments regarding our work, including the novelty of applying JEPA to raw waveforms, the SOTA  empirical performance of our model, and the importance of evaluating audio foundation models in naturalistic scenes.
>
>
> > 1. **Motivation needs to be improved:** There have been some mel-spectrum-based JEPA variants [a, b, c], which are very relevant to WavJEPA. Did you get motivated by them? What's your advantage over them? They should be the baselines in the Experiment section, given they also claim superior performance over baseline models adopted by WavJEPA.
>
>
> Our reviews indicate that a large part of the audio processing community is firmly attached to STFT models. Therefore, we will include STFT baselines in the HEAR and ARCH evaluation suites.
> Naturally, these STFT baselines we will include the STFT-based JEPA models already published.
>
>
> > 2. **Limitations of token/speech unit level:** You mention that the core advantage of WavJEPA is that it "overcomes the limitations of learning representations at the token or speech unit level." (1) What exactly are the "limitations"? I understand there have been some image-based arguments in I-JEPA. However, as your whole work is JEPA on waveform, I would expect in-depth audio-based arguments, paired with qualitative/quantitative analysis on the difference in features learned by WavJEPA and token/unit level methods, instead of only showing that you achieve SOTA performance on benchmarks.
>
>
> We thank the reviewer for pointing out this lack of clarity. We will make sure to better explain this aspect in the next manuscript iteration, and provide more in-depth audio-based arguments.
>
>
> > 3. **Introduction to JEPA:** The introduction to JEPA in 3rd paragraph of the Related Work Section is a bit superficial. It does not mention JEPA's intuition and designs to achieve the goal. You may also introduce some terminology used by the original paper (e.g., "context" and "target"). Now this paragraph only mentions its good performance ("efficiently learn semantic image representations"). Insufficient introduction to JEPA also undermines the readability of the Methodology section.
>
>
> We thank the reviewer for pointing out this gap in our explanation of JEPA. We will make sure to provide a more comprehensive introduction to JEPA in the next manuscript iteration.
>
>
> > 4. **Novelty:** The novelty (settings, engineering techniques, and insights) of this work seems a bit limited. I would assume engineering techniques (e.g., how to adapt JEPA to the waveform) to be the highlights. While the Introduction, Related Work, and Methodology sections do not show this.
>
>
> Our work constitutes the first general purpose foundation model trained on raw waveforms. This is a significant step forward for the audio processing community as it overcomes the limitations of STFT-based methods we described in the article and, in more details, in our global rebuttal.
> Additionally, it is also the first time JEPA is applied to raw waveforms.
> Nevertheless, we understand the reviewer's concerns regarding novelty. We will make sure to better highlight the novelty of our work in the next manuscript iteration.
>
>
>
>
> > 5. **Limitation is not discussed.**
>
>
> We thank the reviewer for pointing out this omission. We will make sure to include a discussion regarding the limitations of our work in the next manuscript iteration.

---

### Official Review · Reviewer_hJKU · 2025-10-27

**Soundness:** 1
**Presentation:** 3
**Contribution:** 1
**Rating:** 2
**Confidence:** 5

**Summary:**

## Learning self-supervised audio representations from raw waveforms using WavJEPA

- The paper proposes WavJEPA, an extension to the JEPA approach for learning self-supervised audio representations from raw waveforms.
- The authors conduct an in-depth evaluation of the proposed approach on several downstream audio recognition tasks.

**Strengths:**

The following are the key strengths of the paper:

- The paper is well written for the most part (see Weakness 1 for problems) and is easy to ingest.
- To the best of my knowledge, this paper is probably the first to pretrain Wav2Vec 2.0 and HuBERT, two seminal SSL models on AudioSet. This evaluation would be useful for the audio SSL community, especially if the models are released publicly.

**Weaknesses:**

The following are the main issues with the paper, in no order of importance.

## W1: Inconsistent writing
The writing is inconsistent, especially with respect to referencing specific SSL models. For instance, in some places it's Wav2vec2.0, while in others it's Wav2Vec 2.0, and in other places it is W2V2.

---

## W2: Choice of evaluated downstream tasks

- The inclusion of NatHEAR appears unmotivated. It felt sudden, like "hey, so we are also including results on this task". What are "Naturalistic" scenes, and why are they of relevance to this paper? This one question is not justified in the paper. I understand the usability of NatHEAR as an independent evaluation benchmark for spatial sound scenes. But NatHEAR comprises the same tasks as the subset of HEAR already evaluated in the paper, albeit spatialized, so it is unclear to me what value the inclusion of this benchmark adds. This is also echoed by the results from Table 3, where, for the most part, the models follow the exact same performance trend between HEAR and NatHEAR.

- While there are legitimate reasons to do so, the justification of using the same subset of HEAR tasks as Yadav et al. (2025) is not clear to me in this instance. If you had to pick a subset of tasks from HEAR, why pick the ones that are already present in ARCH (FSD50K and ESC-50)? If the objective is to limit the total number of evaluations, which was Yadav et al.'s main argument, it doesn't make sense to then redo evaluation on these two tasks.

---

## W3: Weak baselines/No comparison with spectrogram-based audio SSL models

- Comparison with spectrogram-based SSL models, such as BEATs, AudioMAE/MSM/Dasheng, is completely omitted.
- All the models evaluated were *specifically designed for modelling speech*. Their inclusion as crucial baselines in literature makes sense, but they were never designed to learn general audio representations. This is evident from Table 1, where the HEAR-Naive model outperforms all baseline models for 3/4 music-based tasks. Spectrogram-based SSL models fare significantly better than all evaluated baselines for learning *audio* representations. There also is a lack of commentary on justification of WHY these baselines are included.
-  Even if we ignore other spectrogram-based SSL models, as there are a lot of them, the paper also doesn't even draw direct comparisons with spectrogram-based JEPA-style models [1, 2]. ([2] was submitted in June 2025, so we can consider it a concurrent work.)
- I understand that the main objective of the paper is modelling from raw waveforms, but how can we contextualise the performance of the proposed approach if there are no comparisons to the best-performing models for the domain?
- There is merit to proposing a new approach of sufficient novelty and analysis, even if it doesn't become the coveted state-of-the-art, but that is not the case here.

---

## W4: Weak novelty

- The novelty in this paper is in the application, that too only for a specific type of modeling (raw-waveforms). The JEPA architecture is well established, and JEPA based audio representation learning models also exist.

---

### References
1. Fei et al. "A-JEPA: Joint-Embedding Predictive Architecture Can Listen", 2024.
2. Tuncay et al. "Audio-JEPA: Joint-Embedding Predictive Architecture for Audio Representation Learning", 2025. (Submitted to arxiv on June 2025)

**Questions:**

1. The authors mention using the same datasets as Yadav et al. 2024, which include the NSynth-5h (NS-5h) and Speech Commands-5h (SC-5h) tasks instead of the longer, full training set versions (NS and SC, respectively). However, while all the tables explicitly specify SC-5h, they only specify NS and not NS-5h. Is this a typo? If training tasks were based on the decisions made by Yadav et al., why was full NS used?

---

> ### Author Response · Authors · 2025-12-03
>
> We thank the reviewer for the compliments regarding the writing and the novelty of our work. We will now address point by point the concerns.
>
> > 1. **Inconsistent writing:**  The writing is inconsistent, especially with respect to referencing specific SSL models. For instance, in some places it's Wav2vec2.0, while in others it's Wav2Vec 2.0, and in other places it is W2V2.
>
>
> We thank the reviewer for pointing out this naming inconsistency. We will make sure to fix it and check for other naming mistakes in the next iteration.
>
>
> > 2. **Choice of evaluated downstream tasks:**
> > - The inclusion of NatHEAR appears unmotivated. It felt sudden, like "hey, so we are also including results on this task". What are "Naturalistic" scenes, and why are they of relevance to this paper? This one question is not justified in the paper. I understand the usability of NatHEAR as an independent evaluation benchmark for spatial sound scenes. But NatHEAR comprises the same tasks as the subset of HEAR already evaluated in the paper, albeit spatialized, so it is unclear to me what value the inclusion of this benchmark adds. This is also echoed by the results from Table 3, where, for the most part, the models follow the exact same performance trend between HEAR and NatHEAR.
> > - While there are legitimate reasons to do so, the justification of using the same subset of HEAR tasks as Yadav et al. (2025) is not clear to me in this instance. If you had to pick a subset of tasks from HEAR, why pick the ones that are already present in ARCH (FSD50K and ESC-50)? If the objective is to limit the total number of evaluations, which was Yadav et al.'s main argument, it doesn't make sense to then redo evaluation on these two tasks.
>
>
> We thank the reviewer for pointing out the lack of arguments for motivating the evaluation of models in naturalistic scenes. We will make sure to clarify the explanation in the next manuscript iteration.
>
>
> > 3. **Weak baselines/No comparison with spectrogram-based audio SSL models:**
> > - Comparison with spectrogram-based SSL models, such as BEATs, AudioMAE/MSM/Dasheng, is completely omitted.
> > - All the models evaluated were specifically designed for modelling speech. Their inclusion as crucial baselines in literature makes sense, but they were never designed to learn general audio representations. This is evident from Table 1, where the HEAR-Naive model outperforms all baseline models for 3/4 music-based tasks. Spectrogram-based SSL models fare significantly better than all evaluated baselines for learning audio representations. There also is a lack of commentary on justification of WHY these baselines are included.
> > - Even if we ignore other spectrogram-based SSL models, as there are a lot of them, the paper also doesn't even draw direct comparisons with spectrogram-based JEPA-style models [1, 2].
> > - I understand that the main objective of the paper is modelling from raw waveforms, but how can we contextualise the performance of the proposed approach if there are no comparisons to the best-performing models for the domain?
> > - There is merit to proposing a new approach of sufficient novelty and analysis, even if it doesn't become the coveted state-of-the-art, but that is not the case here.
>
>
> As explained in the article and, in more detail, in our global rebuttal, STFT suffers inherent limitations that prevent its use in specific applications. Nevertheless, our reviews indicate that a large part of the audio processing community is firmly attached to STFT models. Therefore, we will include STFT baselines in the HEAR and ARCH evaluation suites.
> Naturally, we will also include the STFT-based JEPA baselines along with the other STFT baselines in our next manuscript iteration.
>
>
> > 4. **Weak novelty:** The novelty in this paper is in the application, that too only for a specific type of modeling (raw-waveforms). The JEPA architecture is well established, and JEPA based audio representation learning models also exist.
>
>
> Our work constitutes the first general-purpose foundation model trained on raw waveforms. This is a significant step forward for the audio processing community as it overcomes the limitations of STFT-based methods we described in the article and, in more details, in our global rebuttal.
> Additionally, it is also the first time JEPA is applied to raw waveforms.
> Nevertheless, we understand the reviewer's concerns regarding novelty. We will make sure to better highlight the novelty of our work in the next manuscript iteration.

---

### Official Review · Reviewer_grr7 · 2025-10-29

**Soundness:** 2
**Presentation:** 2
**Contribution:** 2
**Rating:** 2
**Confidence:** 4

**Summary:**

The paper proposes WavJEPA, a waveform-domain adaptation of the Joint-Embedding Predictive Architecture (JEPA) for general-purpose audio representation learning. The authors also introduce WavJEPA-Nat, a dual-channel variant trained on simulated naturalistic scenes with a 2-D positional scheme to capture inter-channel structure. Across HEAR and ARCH, WavJEPA reports strong gains versus prior time-domain baselines; WavJEPA-Nat further improves robustness on Nat-HEAR. The paper argues that learning from waveforms avoids spectrogram latency and phase loss, pointing to potential low-latency applications, though no latency measurements are provided.

**Strengths:**

1. **Empirical performance (time-domain):** Consistent improvements over wav2vec2 / HuBERT / WavLM / Data2Vec on HEAR and ARCH with a 90 M-parameter model. The scaling plot (Fig. 2) is informative.

2. **Robustness in naturalistic scenes:** WavJEPA-Nat outperforms other time-domain baselines on Nat-HEAR and shows a smaller performance drop from HEAR to Nat-HEAR.

3. **Clear methodology:** The sampling scheme, EMA target encoder, and top-K target averaging are described and ablated.

4. **Broad downstream coverage:** Evaluations span events/scenes, speech, and music, with per-task breakdowns that help diagnose where the method shines.

**Weaknesses:**

# 1. Missing baselines
The paper does not compare to strong spectrogram-based self-supervised models (e.g., MAE-style, AST/SSAST, BEATs/EAT) despite emphasizing spectrogram limitations in the introduction and reviewing these methods in related work. This makes it hard to judge the true SOTA position and whether gains are due to modality (waveform) vs. training recipe (JEPA). Please add direct comparisons using the authors' HEAR/ARCH protocols.

Baselines that should be included:
- data2vec (AudioSet version, recipe: https://github.com/facebookresearch/fairseq/blob/main/examples/data2vec/config/audio/pretraining/audioset.yaml)
- data2vec 2.0
- SSAST (https://arxiv.org/abs/2110.09784)
- BEATs (https://arxiv.org/abs/2212.09058)
- ATST-Frame (https://arxiv.org/abs/2306.04186)

Strong baselines that are trained with more data:
- Dasheng (https://arxiv.org/abs/2406.06992)
- USAD (https://arxiv.org/abs/2506.18843)

# 2. Latency
The abstract/intro repeatedly claim that waveform learning avoids STFT latency and enables "low-latency" deployment, but there are no end-to-end latency or throughput measurements. Moreover, the latency introduced by the waveform encoder is a known issue with wav2vec2-style SSL models. Please quantify end-to-end latency and compare against an optimized spectrogram pipeline (e.g., streaming STFT).

# 3. Temporal resolution
WavJEPA uses a 100 Hz frame rate after removing the final convolutional layer, while the waveform baselines use 50 Hz. Without an ablation that matches the frame rate, some gains may be due to higher temporal resolution rather than JEPA training. Please include a feature-rate ablation (retain the last convolutional layer or adjust strides to 50 Hz).

# 4. Computational costs
The paper claims efficiency ("fraction of the data" and "fewer computational resources") but does not report training compute (FLOPs) or GPU-days for all models under comparison. Include standardized compute metrics and data seen for each curve/point in Fig. 2 and for baselines where feasible.

# 5. Presentation issues
Figures are sometimes far from the relevant text, so try placing them at the top of pages and before the first reference in the text.
In Fig. 3, the middle and right subfigures plot per-task values as lines over categorical x-axes (different tasks). Use grouped bar charts rather than curves to avoid implying continuity or correlation across tasks.

**Questions:**

1. Add spectrogram-baseline comparisons on HEAR, ARCH, and Nat-HEAR with the same frozen-encoder linear-probe protocol used here. If infeasible for all, include at least one strong spectrogram foundation model per family (MAE-style, AST/SSAST, BEATs/EAT).

2. Latency: Provide end-to-end streaming latency and real-time factor for (i) spectrogram baseline and (ii) WavJEPA pipeline (CNN + encoder), measured on the same hardware. Also report receptive-field/algorithmic latency for each stage.

3. Temporal resolution ablation: Re-train WavJEPA with a 50 Hz frame rate (restore final conv or adjust stride) and report HEAR/ARCH deltas to isolate the effect of feature rate vs. JEPA objective.

4. Compute reporting: For Fig. 2 and headline results, report data-seen, optimizer steps, batch sizes, GPU-days, and estimated training FLOPs for each point/model; clarify whether competing baselines' numbers come from your re-runs or prior work.

5. Nat-HEAR fairness: Add a control where WavJEPA-Nat processes single-channel input (or duplicated channels) to quantify the benefit from dual-channel modeling. Conversely, where feasible, provide a two-channel baseline.

6. $s(m)$ metric: This metric was proposed here
> Feng, Tzu-hsun, et al. "Superb@ slt 2022: Challenge on generalization and efficiency of self-supervised speech representation learning." 2022 IEEE Spoken Language Technology Workshop (SLT). IEEE, 2023.

---

> ### Author Response · Authors · 2025-12-03
>
> We appreciate the reviewer's thoughtful and encouraging assessment of our work, particularly regarding empirical performance, robustness in naturalistic scenes, clear methodology, and broad downstream coverage.
> Below are our individual responses to the points raised.
>
> > 1. **Missing baselines:** The paper does not compare to strong spectrogram-based self-supervised models (e.g., MAE-style, AST/SSAST, BEATs/EAT) despite emphasizing spectrogram limitations in the introduction and reviewing these methods in related work. This makes it hard to judge the true SOTA position and whether gains are due to modality (waveform) vs. training recipe (JEPA). Please add direct comparisons using the authors' HEAR/ARCH protocols.
> Baselines that should be included: data2vec (AudioSet version), data2vec 2.0, SSAST, BEATs, ATST-Frame, Dasheng, USAD.
>
> As explained in the article and, in more detail, in our global rebuttal, STFT suffers inherent limitations that prevent its use in specific applications.
> Nevertheless, our reviews indicate that a large part of the audio processing community is firmly attached to STFT models. Therefore, we will include STFT baselines in the HEAR and ARCH evaluation suites.
>
> > 2. **Latency:** The abstract/intro repeatedly claim that waveform learning avoids STFT latency and enables "low-latency" deployment, but there are no end-to-end latency or throughput measurements. Moreover, the latency introduced by the waveform encoder is a known issue with wav2vec2-style SSL models. Please quantify end-to-end latency and compare against an optimized spectrogram pipeline (e.g., streaming STFT).
>
> While STFT overhead is well-established in audio processing, we agree that explicit quantification strengthens our contribution. Therefore, the revised manuscript will include comprehensive latency benchmarks comparing STFT-based and raw waveform approaches.
>
> > 3. **Temporal resolution:** WavJEPA uses a 100 Hz frame rate after removing the final convolutional layer, while the waveform baselines use 50 Hz. Without an ablation that matches the frame rate, some gains may be due to higher temporal resolution rather than JEPA training. Please include a feature-rate ablation (retain the last convolutional layer or adjust strides to 50 Hz).
>
> Thank you for pointing out this difference with other waveform baselines. We will run 50 Hz WavJEPA experiments and include them as an ablation study.
>
> > 4. **Computational costs:** The paper claims efficiency ("fraction of the data" and "fewer computational resources") but does not report training compute (FLOPs) or GPU-days for all models under comparison. Include standardized compute metrics and data seen for each curve/point in Fig. 2 and for baselines where feasible.
>
> Our WavJEPA framework uses a similar transformer architecture as the other waveform baselines. Therefore, the number of samples seen (Figure 2) can be used to approximate the FLOPs or GPU-days. Moreover, the difference in the amount of data required to train WavJEPA compared to its competitors is substantial (1/10 and nearly 1/100 of Wav2Vec2.0 and HuBERT, respectively), which supports this claim confidently.
> Nevertheless, we will compute and provide the FLOP estimations for completeness, and we thank the reviewer for pointing out this gap.
>
> > 5. **Presentation issues:** Figures are sometimes far from the relevant text, so try placing them at the top of pages and before the first reference in the text. In Fig. 3, the middle and right subfigures plot per-task values as lines over categorical x-axes (different tasks). Use grouped bar charts rather than curves to avoid implying continuity or correlation across tasks.
>
> We thank the reviewer for providing these suggestions for improvement. We will make sure to include them in the following manuscript iteration.

---

### Official Review · Reviewer_Q6ym · 2025-11-03

**Soundness:** 3
**Presentation:** 2
**Contribution:** 2
**Rating:** 2
**Confidence:** 3

**Summary:**

This paper leverages self-supervised learning based on the Joint-Embedding Predictive Architecture (JEPA) for waveforms to obtain robust semantic representations from audio data. The method introduces a novel masking strategy designed to capture semantic information more effectively than previous masking-based approaches such as HuBERT and Data2Vec, although the details of this improvement are somewhat difficult to fully grasp. The approach is further extended to a multichannel version (WavJEPA), which enhances robustness against noise and reverberation. Experiments are conducted across three sound domains—speech, music, and environmental sounds—and demonstrate the effectiveness of the proposed methods when applied to popular speech SSL models pre-trained on either speech or AudioSet datasets.

**Strengths:**

- Application of JEPA to SSL for waveform audio, enabling robust semantic representation learning.
- Proposed masking strategy that effectively captures richer audio semantics compared to previous masking-based approaches.
- Comprehensive evaluation across speech, music, and environmental sound domains, including thorough ablation studies to validate design choices.

**Weaknesses:**

* **Mismatch of method choices:** While the paper includes speech, its goal is to learn general audio representations. However, the comparisons are limited to speech-oriented SSL models (Wav2Vec 2.0, HuBERT, Data2Vec), which are primarily designed for speech tasks. Simply changing their training data does not make them fully representative of general audio. Comparisons with general-purpose audio encoders (e.g., PANN, AST, BEATS, Dashen) would make the evaluation more appropriate.
* **Evaluation against established benchmarks:** The method should be evaluated using standard metrics from the HEAR benchmark, and the experimental design could be better aligned with HEAR to allow more direct comparisons.
* **Unclear masking strategy:** The procedure for sampling context and target blocks is difficult to follow. Figure 1 does not fully clarify the process, and a dedicated figure or algorithm-style pseudocode would improve comprehension.
* **Limited novelty:** JEPA has already been applied to the audio domain (e.g., Self-Supervised Representation Learning with a JEPA Framework for Multi-instrument Music Transcription in https://waspaa.com/author-index/). Therefore, this work is not the first application of JEPA to audio, and the novelty of the approach is relatively limited.

**Questions:**

1. **Clarification on context and target block sampling:**
   The explanation of sampling the context and target blocks is confusing, particularly due to multiple sampling strategies and iterative procedures. It would greatly improve clarity if a figure or algorithm-style pseudocode were provided to illustrate the process.

2. **Notation for top $K$ layers and $K_{\text{target}}$:**
   The use of the same symbol $K$ for both top layers and $K_{\text{target}}$ is confusing. Could the authors clarify the distinction or consider using separate notation to avoid ambiguity?

3. **Table readability:**
   Tables 1, 2, and 3 are quite small and difficult to read. Increasing their size or improving formatting would enhance readability and accessibility of the results.

---

> ### Author Response · Authors · 2025-12-03
>
> We thank the reviewer for their detailed analysis and constructive feedback. In particular, we appreciate the recognition of our method's strengths in applying JEPA to waveform audio, our masking strategy, and the breadth of our evaluations.
> Below are responses to the individual points and the modifications we plan to apply before resubmitting in another venue.
>
> > **Mismatch of method choices:** While the paper includes speech, its goal is to learn general audio representations. However, the comparisons are limited to speech-oriented SSL models (Wav2Vec 2.0, HuBERT, Data2Vec), which are primarily designed for speech tasks. Simply changing their training data does not make them fully representative of general audio. Comparisons with general-purpose audio encoders (e.g., PANN, AST, BEATS, Dashen) would make the evaluation more appropriate.
>
> At the moment, the only time-domain foundation models were explicitly developed for speech, and there are no foundation models for general-purpose audio representation learning at the moment. This is precisely the gap we are aiming to fill with this work.
> Nevertheless, our reviews indicate that a large part of the audio processing community is firmly attached to STFT models. Therefore, despite their inherent limitations, as described in the article and, in more detail, in our global rebuttal, we will include STFT baselines in the HEAR and ARCH evaluation suites.
>
> > **Evaluation against established benchmarks:** The method should be evaluated using standard metrics from the HEAR benchmark, and the experimental design could be better aligned with HEAR to allow more direct comparisons.
>
> The results we provided already include the HEAR metrics. We additionally offered an aggregated metric s(m) which was introduced in SUPERB (Yang et al., 2021) to have a global score for each method. The confusion might come from the fact that s(m) was not present in the original HEAR benchmark.
> We will make this subtlety more straightforward in the following manuscript iteration.
>
> > **Unclear masking strategy:** The procedure for sampling context and target blocks is difficult to follow. Figure 1 does not fully clarify the process, and a dedicated figure or algorithm-style pseudocode would improve comprehension.
>
> Thank you for pointing this out. We will add pseudocode for the masking process to improve clarity.
>
> > **Limited novelty:** JEPA has already been applied to the audio domain (e.g., Self-Supervised Representation Learning with a JEPA Framework for Multi-instrument Music Transcription in https://waspaa.com/author-index/). Therefore, this work is not the first application of JEPA to audio, and the novelty of the approach is relatively limited.
>
> To our knowledge, the existing applications of JEPA to audio all relied on STFT representations. Our work constitutes the first application of JEPA to raw waveforms. Naturally, we will include these STFT-based JEPA baselines along with the other STFT baselines in our next manuscript iteration.

---

### Author Response · Authors · 2025-12-03

We would like to thank the reviewers for their thoughtful assessment of our submission to ICLR 2026. In the following, we provide additional context and clarifications that address the main concerns raised during the review process.

**Limitations of spectrogram models:**
Spectrogram models use only the magnitude information from the Short-Time Fourier Transform (STFT), discarding the phase information. To obtain a waveform from a spectrogram, algorithms such as Griffin-Lim or vocoders are needed to approximate it. Consequently, generative tasks with spectrogram-based foundation models are of limited quality and are bounded by the quality of the waveform approximation. In contrast, time-domain models preserve the full waveform and are not subject to this upper bound..

Furthermore, achieving high-resolution frequency decomposition with STFT necessitates the use of relatively long analysis windows (e.g., 25 ms in recent foundation models). These windows introduce latency in real-time applications, as enough samples must be accumulated before computing the transform. Time-domain approaches such as WavJEPA avoid this limitation: their effective context is governed by the receptive field of the convolutional architecture, which is typically shorter and offers substantially greater flexibility.

Together, these considerations make STFT-based models suboptimal for real-time inference and for high-quality generative audio tasks. This explanation is intended to clarify the rationale underlying our decision not to include STFT-based baselines in the benchmark. Nevertheless, we recognize from the reviews that a substantial portion of the audio processing community continues to rely heavily on STFT representations. In light of this, and despite their limitations, we will incorporate STFT baselines into the HEAR and ARCH evaluation suites to facilitate broader comparison.

**Novelty of JEPA:**
Our work introduces the first general-purpose foundation model trained directly on raw waveforms, marking an important milestone for the audio community by addressing the limitations of spectrogram-based methods. Additionally, this is—to the best of our knowledge—the first application of the Joint-Embedding Predictive Architecture (JEPA) to raw audio signals.

**Retraining baselines:**
Finally, we note that the data2vec model trained on AudioSet and the data2vec2.0 models are not available through HuggingFace or seq2seq repositories. We wish to emphasize the importance of open science and reproducibility: as an academic institution, we do not have the computational resources required to retrain data2vec or data2vec2.0 on large-scale datasets such as AudioSet. For foundation models, releasing only the training code is insufficient; public availability of model weights is essential for meaningful comparison and reproducible research.

---

### Note · Authors · 2025-12-03

**Comment:**

Considering the low grades our article received, we decided to withdraw it from ICLR, in order to avoid wasting the reviewers' and area chair's time with another round of rebuttal and review.

Nevertheless, we believe in the strength of our work and therefore provided detailed responses to all reviewers.

**Withdrawal Confirmation:**

I have read and agree with the venue's withdrawal policy on behalf of myself and my co-authors.